# Revisiting fairGNN-WOD: A Reproduction and Analysis of Fair Graph Learning Without Demographics

## Abstract

Graph Neural Networks are widely used for high-impact prediction tasks where fairness is important. fairGNN-WOD (Wang et al., 2025b) proposes a two-stage framework to make fair predictions without relying on sensitive demographics information, which is often unavailable. In this study, we implement fairGNN-WOD[1] from scratch due to the lack of a publicly available code base. Ultimately, while we reproduce utility results, we fail to reproduce the reported fairness improvements, which is the main contribution of the original paper, because baselines are substantially more fair than originally reported. Additionally, we find no measurable contribution of stage 1 of the original framework, which is a key architectural component of fairGNN-WOD.

## 1 Introduction

Graph Neural Networks (GNNs) have found widespread use in high-stakes decision-making domains such as credit risk assessment, hiring, and recommendation systems. For such tasks, it is important that models are not only accurate, but also fair, and balancing these requirements is not trivial. Existing methods often mask sensitive information, which can reduce utility because sensitive attributes may contain task-relevant information, or use sensitive information to account and correct for the bias present in a model. While the latter of these approaches can improve fairness while maintaining utility, it depends on the availability of sensitive information, which, in practice, is often absent in datasets due to privacy concerns.

FairGNN-WOD (Wang et al., 2025b) proposes to address this problem by introducing a new two-stage framework that does not rely on sensitive demographics information. In the first stage, a variational graph autoencoder (VGAE) (Kipf & Welling, 2016) is used to infer missing sensitive demographics information by learning a latent representation that is used to reconstruct the graph structure and node features. In the second stage, a graph neural network (GNN) is used to construct and disentangle node representations into task-relevant components and demographics-relevant components that are subsequently masked. A key innovation is that fairGNN-WOD disentangles sensitive information such that task-relevant information is preserved while bias-inducing information is suppressed. The authors claim that fairGNN-WOD achieves state-of-the-art fairness while maintaining comparable utility, without the use of sensitive demographics information.

This study implements the fairGNN-WOD framework proposed by Wang et al. (2025b) from scratch and evaluates the claims made in the original paper. We reproduce the reported utility but not the reported fairness improvements, which we trace to baselines that are already near-fair in our setup and to a reported baseline disparity we could not reconstruct under any tested configuration. Additionally, we catalogue the key ambiguities that complicated reproduction in Appendix A.

Finally, we extend the study to probe the method's central claims directly by substituting the first-stage generative model with simpler proxies and a random-noise baseline, and by varying the fraction of demographic supervision. We find that the first stage has no measurable effect on downstream fairness and that the method improves fairness only on datasets where sensitive attributes meaningfully correlate with the target label.

---

[1]Our code is publicly available at `https://anonymous.4open.science/r/fairgnn-wod-extension-CD93/`

## 2 Scope of Reproducibility

This work aims to reproduce the findings of Wang et al. (2025b) about the performance of fairGNN-WOD, and provides insight on the contribution of Stage 1 of the framework and the role of sensitive demographics during training. Specifically:

- We aim to verify the original claim that fairGNN-WOD significantly reduces bias compared to GNN baselines.

- We aim to verify the original claim that fairGNN-WOD successfully navigates the fairness-utility trade-off and maintains utility compared to unconstrained baselines.

- We investigate the necessity of the Stage-1 VGAE by substituting it with simpler demographic proxies and a random noise baseline, testing whether the generative architecture provides any measurable benefit over alternatives that require no graph reconstruction.

- We test whether Stage-1 requires any demographic supervision to be effective, by varying the fraction of sensitive label visible during training from 0% to 100% and measuring the downstream effect on fairness.

- We use three additional datasets, increasing the robustness of our findings.

Due to the original codebase being unavailable, implementation of the framework is done from scratch based on the description given in the original paper, which is sufficiently ambiguous to allow for some discrepancies between our implementation of the framework and the original implementation.

Ultimately, we are unable to reproduce the findings made by Wang et al. (2025b). This is, at least in part, likely due to the unavailability of the original codebase and ambiguities in the original description.

## 3 Methodology

We describe our implementation of the fairGNN-WOD framework and our experimental setup. We explicitly state what is clear in the original description by Wang et al. (2025b) and what design choices are based on our informed interpretation of the original work where it is vague.

### 3.1 Model Description

The fairGNN-WOD framework consists of two independently trained stages. Stage 1 consists of a VGAE that reconstructs graphs from a latent space that represents sensitive demographics information $S$. Stage 2 uses $S$ from Stage 1 to disentangle node representations into task-relevant and demographics-relevant components, the latter of which are subsequently masked. Training is decoupled into two sequential stages: Stage 1 is trained first, then frozen, with its reconstruction of $S$ serving as ground truth demographics for the training of Stage 2. Our final implementation of the framework is depicted in Figure D.1, Appendix D.

### 3.1.1 Stage 1: VGAE

Wang et al. (2025b) describe that a VGAE is employed in the first stage of the model to infer missing demographics information that can subsequently be used by the second stage of the model. To justify the utility of a VGAE as Stage 1 the authors posit that "the estimates of the latent variable representing the true demographic information are consistent with the underlying data generation process" (Wang et al., 2025b), and that "by learning this process, we can derive a latent representation $Z$ that encapsulates as much information about the true demographic information $S$ as possible".

Specifically, the paper describes that the VGAE estimates the joint distribution $P(Z, S, X, A)$, where $Z$ denotes a latent representation, $S$ corresponds to missing demographic information, and $X$ and $A$ represent the node features and graph structure, respectively. This joint distribution is factorised into

$P(Z)P(S|Z)P(A|S)P(X|A,S)$, where $P(Z)$ is a unit-Gaussian prior of latent representation $Z$. As described by Wang et al. (2025b), the distributions $P(Z|A,X)$, $P(S|Z)$, $P(A|S)$ and $P(X|A,S)$ are modelled using a VGAE and enable the inference of $S$. Moreover, the paper describes the VGAE as consisting of two encoders that approximate $P(Z|A,X)$ and $P(S|Z)$ with variational distributions $q_\phi(Z|A,X)$ and $q_\psi(S|Z)$, respectively. To model $P(A|S)$ and $P(X|A,S)$, two decoders are employed that produce the node features and graph structure.

**In our experiments**, we implement the VGAE with two encoders in sequence that model $P(Z|A,X)$ and $P(S|Z)$, referred to as $q_\phi$ and $q_\psi$, respectively. $q_\phi(Z|A,X)$ parameterises a normal distribution, and $q_\psi(S|Z)$ models a relaxed categorical distribution, where $S \in \mathbb{R}^C$ represents the continuous relaxation of $C$ distinct demographic groups encoded in the latent space. The encoders are followed by two parallel decoders that model $P(A'|S)$ and $P(X'|A,S)$, referred to as $p_\theta$ and $p_\eta$, respectively.

In our experiments, $q_\psi$ and $p_\theta$ are implemented as a three- and two-layer MLP, respectively, with LeakyReLU activations after all except the final layer. $q_\phi$ and $p_\eta$ are implemented as a two- and one-layer GCN, respectively, followed by a single fully connected layer and using ReLU activations after its convolutional layers.

To achieve fair demographic predictions, the latent representations modelled by the VGAE must be independent of the downstream classification labels $Y$. To this end, **the original paper** includes the Hirschfeld-Gebelein-Renyi (HGR) maximal correlation (Gebelein, 1941) in the optimisation objective to penalise dependence between $Z$ and $Y$:

$$\text{HGR}(Z,Y) = \sup_{p_Z, p_Y} \frac{\mathbb{E}[p_Z(Z) \cdot p_Y(Y)]}{\sqrt{\mathbb{E}[p_Z^2(Z)] \cdot \mathbb{E}[p_Y^2(Y)]}} \tag{1}$$

The original paper describes $p_Z$ and $p_Y$ as approximated via two interconnected neural networks. **In our experiments**, $p_Z$ and $p_Y$ are modelled via a combined two-layer MLP with LeakyReLU activations after the first layer. To provide the correlation estimator with sufficient discriminative capacity, this MLP is designed to be significantly wider than the latent space.

Finally, **the original paper** optimises the VGAE by maximizing the Evidence Lower Bound (ELBO) (Kingma & Welling, 2022) of the marginal data likelihood combined with the HGR maximal correlation scaled by a factor $\lambda$:

$$\begin{aligned} \log P(X,A) \geq \\ \mathbb{E}_{q_\phi(Z|A,X)}[\mathbb{E}_{q_\psi(S|Z)}[\log P(X|S,A) + \log P(A|S)]] \\ - KL(q_\psi(S|Z)||P(S|Z)) - KL(q_\phi(Z|A,X)||P(Z)) - \lambda \cdot \text{HGR}(Z,Y) \end{aligned} \tag{2}$$

where $\log P(X|S,A)$ and $\log P(A|S)$ represent the reconstruction losses for the node features and graph structure. **In our experiments**, we use standard mean squared error (MSE) for reconstruction losses, following related work (Wang et al., 2025a). The VGAE hyperparameters are specified in Table C, Appendix C.

### 3.1.2 Stage 2: Disentangled Fair GNN

According to Wang et al. (2025b), in the second stage, a GNN leverages the inferred demographic proxies to build unbiased node representations. The original framework outlines an architecture where node embeddings are divided into $N_c$ distinct channels to disentangle latent factors. Wang et al. (2025b) utilise a discriminator to identify which channels capture demographic information, allowing them to be masked to obscure demographic cues while preserving task-related information.

To this end, the authors introduce a linear transformation to obtain initial disentangled node representations from node features and an *adaptive assigner* $F_\psi$ that computes channel-specific neighborhood weights to guide message passing, yielding the following convolutional update rule:

$$h_{c,i}^{(l+1)} = \sigma\left(\sum_{v_j \in \mathcal{N}(i)} \omega_{i,j}^c \cdot h_{c,j}^{(l)} \cdot \mathbf{W}^{c,(l)}\right) \tag{3}$$

where $\mathcal{N}(i)$ denotes the neighbors of node $v_i$, $\mathbf{W}^{c,(l)}$ is the learnable weight matrix for layer l in channel c, $\sigma(\cdot)$ is an activation function and $\omega_{i,j}^c$ is the weight for channel c from node $v_i$ to node $v_j$:

$$\omega_{i,j} = \text{softmax}\left(F_\psi([x_i; x_j])\right) \tag{4}$$

where $[x_i; x_j]$ denotes the concatenation of the features of nodes $v_i$ and $v_j$.

Finally, to identify and mask channels with demographics-relevant information, a *discriminator* is used in combination with a *learnable masking algorithm*. As described in the original paper, the adaptive assigner is implemented as a multilayer perceptron. **In our experiments**, we use a two-layer MLP with a ReLU activation after the first layer. The GNN itself is implemented with two layers, also using a ReLU activation after the first layer.

The discriminator identifies demographics-relevant channels by predicting $S$ from each channel independently. The per-channel sensitivity score is computed as:

$$\text{sens}_c = \left(1 - \frac{\mathcal{L}_D^c}{H(\hat{S})}\right)_+ \tag{5}$$

where $\mathcal{L}_D^c$ is the average cross-entropy loss of predicting $S$ from channel $c$ and is equivalent to Equation 11, and $H(\hat{S})$ is the entropy of the Stage 1 sensitive attribute predictions used as a normalisation. A learnable mask parameter $\mathbf{M} \in \mathbb{R}^{C \times H}$ then produces a soft mask via:

$$\mathbf{m}_c = (1 - \text{sens}_c) \cdot \mathbf{1} + \text{sens}_c \cdot (1 - \sigma(\mathbf{M}_c)) \tag{6}$$

We apply a sensitivity-weighted soft mask to the channel embeddings that dynamically suppresses channels with high demographics leakage:

$$\tilde{h}_{c,i} = h_{c,i} \odot \mathbf{m}_c \tag{7}$$

The discriminator is trained adversarially; specifically, gradient reversal is applied to $h$ before the discriminator during the forward pass. During backpropagation, gradients are multiplied by $-\lambda_{DD}$, which forces the disentangled encoder to suppress features the discriminator exploits to predict $S$.

To encourage maximal independence between channels, i.e. diversity such that no overlapping latent factors are embedded in different channels, **the original paper** describes that an Independence Constraint using the Maximum Mean Discrepancy (MMD) (Gretton et al., 2006) is used, where the MMD between two channels is defined as:

$$MMD^2(k_1, k_2) = \frac{1}{n^2}\sum_{i=1}^{n}\sum_{j=1}^{n} f(k_{1,i}, k_{1,j}) + \frac{1}{n^2}\sum_{i=1}^{n}\sum_{j=1}^{n} f(k_{2,i}, k_{2,j}) - \frac{2}{n^2}\sum_{i=1}^{n}\sum_{j=1}^{n} f(k_{1,i}, k_{2,j}) \tag{8}$$

where $k_c$ corresponds to the channel-$c$ representations gathered across all nodes and $f(\cdot, \cdot)$ is a kernel function. Then, the Independence Constraint for *all* channels is:

$$\mathcal{L}_I = \sum_{i=1}^{N_c}\sum_{j=i+1}^{N_c} MMD^2(k_i, k_j) \tag{9}$$

**In our experiments**, $\mathcal{L}_I$ is implemented with the Gaussian RBF kernel, which is mentioned as the only example in the original paper.

**In the original paper**, the learnable masking algorithm is encouraged to effectively eliminate demographics cues from node representations. To this end, another loss term is introduced that penalises the absolute covariance between the obfuscated demographic attribute and the predicted demographics label:

$$\mathcal{L}_F = |\text{Cov}(S, \hat{y})| = |\mathbb{E}[(S - \mathbb{E}(S))(\hat{y} - \mathbb{E}(\hat{y}))]| \tag{10}$$

To encourage the discriminator to effectively learn to predict demographics labels, a classification loss is also introduced:

$$\mathcal{L}_D = \frac{1}{|V_L|}\sum_{v_i \in V_L}\sum_{c=1}^{N_c} [y_{s_i}\log(\hat{y}_{s_i,c}) + (1 - y_{s_i})\log(1 - \hat{y}_{s_i,c})] \tag{11}$$

where $y_{s_i}$ is the demographic information obtained from Stage 1 for node $v_i$, and $\hat{y}_{s_i,c}$ is the predicted demographics. Lastly, a utility performance loss is defined:

$$\mathcal{L}_P = \frac{1}{|V_L|} \sum_{v_i \in V_L} - [y_i \log(\hat{y}_i) + (1 - y_i) \log(1 - \hat{y}_i)] \tag{12}$$

$\mathcal{L}_I$, $\mathcal{L}_F$, $\mathcal{L}_D$ and $\mathcal{L}_P$ are all implemented in our experiments as described in the original paper. The total minimisation objective becomes:

$$\min \mathcal{L}_{GNN} = \mathcal{L}_P + \alpha \mathcal{L}_I + \beta \mathcal{L}_D + \gamma \mathcal{L}_F \tag{13}$$

where $\alpha$, $\beta$ and $\gamma$ are tunable hyperparameters. Note that the original paper uses $\alpha = \beta$.

## 3.2 Datasets

We evaluate on six publicly available graph datasets: **three used in the original paper (Credit, Pokec-z, Pokec-n)** and three additional datasets (Bail, NBA, DBLP) to broaden the evaluation. Table 3.2 summarises key statistics. All datasets are pre-processed as undirected graphs with binary sensitive attributes and binary target labels.

**The Credit dataset** (Yeh & Lien, 2009) shows defaults and payment records for credit card holders. Nodes represent individuals and edges are established based on similarities in payment behaviour. The sensitive attribute $S$ is the individual's age (binarised at the median) and the target label $Y$ is a binary indicator of whether a user will default on their next payment. **The Pokec-z and Pokec-n datasets** are subsets of a Slovak social network (Takac & Zabovsky, 2012). Nodes represent users from two distinct provinces (Zilina and Nitra) and edges represent friendships. For both datasets, the sensitive attribute is the user's region and the prediction task is a binary work-field classification. **The Bail dataset** (Agarwal et al., 2021) contains records of defendants released on bail in the US. Nodes represent defendants, and edges are constructed based on similarity of criminal history. The sensitive attribute $S$ is race (White / non-White) and the target label $Y$ is whether the defendant was re-arrested during the follow-up period. **The NBA dataset** (Dai & Wang, 2021) is a social network of professional basketball players. Node features include performance statistics and salary; edges represent connections between players. The sensitive attribute $S$ is nationality (US / non-US) and the target label $Y$ is whether a player's salary exceeds the median. **The DBLP dataset** (Weis et al., 2006) is a co-authorship network of academic researchers. Node features are bag-of-words representations of publication titles ($d = 2530$); edges represent co-authorships. The sensitive attribute $S$ is inferred gender and the target label $Y$ is research area. The train split is notably small (2.2% nodes) due to class balancing.

| Dataset | Nodes | Edges | Features | Train / Val / Test | | | Sensitive ($S$) |
|---------|-------|-------|----------|-------|------|------|-----------|
| Credit | 30,000 | 230,526 | 13 | 15,000 | 7,500 | 7,500 | Age |
| Pokec-z | 67,796 | 1,303,712 | 276 | 5,131 | 2,565 | 2,566 | Region |
| Pokec-n | 66,569 | 1,100,663 | 265 | 4,398 | 2,199 | 2,200 | Region |
| Bail | 18,876 | 422,853 | 16 | 9,438 | 4,719 | 4,719 | Race |
| NBA | 403 | 16,973 | 95 | 156 | 78 | 79 | Nationality |
| DBLP | 20,111 | 135,127 | 2,530 | 452 | 382 | 398 | Gender |

Table 1: Statistics of the benchmark datasets. The small DBLP training split reflects a highly imbalanced label distribution paired with stratification on $Y$.

## 3.3 Training Configuration

FairGNN-WOD is trained in two stages. Stage 1 trains the demographic proxy model, which is then frozen. Stage 2 trains the disentangled GNN using the frozen Stage 1 predictions.

### 3.3.1 Stage 1

All Stage 1 proxy models (original VGAE, `LatentClassifierVGAE`, `SClassifier`) are trained with the shared hyperparameters in Table C in Appendix C. For the VGAE variants, dynamic weighting is applied to

the KL and HGR terms to allow reconstruction to stabilise before fairness constraints are imposed. Schedules are detailed in Table C. The `SClassifier` uses no auxiliary losses and is trained with cross-entropy on $S$ labels only.

Early stopping monitors validation loss (VGAE variants) or validation accuracy on $S$ (`SClassifier`) with a patience of 20 epochs and a warmup of 50 epochs. Observed median stopping epochs across datasets are reported in Table 3.3.1.

| Model | Bail | Credit | DBLP | NBA | Pokec-n | Pokec-z |
|---|---|---|---|---|---|---|
| GCN | 123 | 80 | 50 | 53 | 60 | 57 |
| FairKD | 169 | 77 | 55 | 73 | 63 | 69 |
| DemographicVGAE | 63 | 63 | 50 | 56 | 53 | 65 |
| LatentClassifierVGAE | 54 | 61 | 59 | 58 | 67 | 67 |
| SClassifier | 60 | 56 | 50 | 55 | 61 | 75 |
| fairGNN-WOD | 100 | 78 | 66 | 62 | 58 | 55 |

Table 2: Median best-model epoch (early stopping epoch $-$ patience) across five seeds. Models that reached max epochs (200) without early stopping are shown as 200.

### 3.3.2 Stage 2

Stage 2 uses the same shared hyperparameters as Stage 1. To prevent collapse during adversarial training, all Stage 2 loss terms are introduced in a staggered manner via the schedules in Table C. The fairness penalty $\mathcal{L}_F$ is introduced first (epoch $0.2W$, where $W$ is length of warmup in epochs) to prioritise bias mitigation early together with the demographic identification loss $\mathcal{L}_D$ (epoch $0.2W$), and finally the independence constraint $\mathcal{L}_I$ (epoch $0.4W$) after learned representations have stabilised. The gradient reversal strength $\lambda_{\mathrm{DD}}$ is ramped up last via a sigmoid schedule. Masking is disabled for the first 20% of training epochs to allow the discriminator to stabilise before the mask is applied.

Early stopping monitors validation F1 score with patience 20 and warmup 50. Median stopping epochs across datasets are reported in Table 3.3.1 and full hyperparameter details are recorded in Appendix C.

### 3.4 Extensions

In addition to evaluating on three datasets beyond the original paper, we extend the study in two directions aimed at clarifying the role of Stage 1 of the framework and the degree of sensitive demographic supervision the method requires. Henceforth, the Stage 1 model used to infer sensitive demographics $S$ may be referred to as the *Stage 1 proxy* or just *proxy*.

### 3.4.1 Alternative Stage 1 Proxies

A central claim of Wang et al. (2025b) is that the VGAE's generative structure by reconstructing $A$ and $X$ from a latent space representing $S$ produces a demographic proxy superior to simpler approaches. We test this claim by substituting the original VGAE with two alternative Stage 1 models:

**SClassifier.** A two-layer GCN trained directly to predict $S$ via cross-entropy. This is a purely discriminative baseline with no graph generative modelling or regularisation. It represents the simplest possible demographic proxy.

**LatentClassifierVGAE.** A VGAE in which the graph is reconstructed from the latent $Z$ directly, and $S$ is predicted as a separate classification head on $Z$. This separates the question of whether reconstruction quality truly depends on the causal generative process with $S$ as a latent space, as the original authors claim.

**Random Noise.** Even at $\rho = 0\%$, Stage 1 models utilise graph structure and may retain weak correlations with the true demographic signal. To eliminate this, we introduce a zero-information baseline in which sensitive attribute predictions are sampled randomly as $\hat{s}_i \sim \text{Bernoulli}(0.5) \; \forall i$. Any downstream fairness

improvement over this baseline must stem from genuine demographic signal in stage 1 rather than from the $\mathcal{L}_F$ covariance penalty acting on arbitrary inputs.

With the exception of Random Noise, all Stage 1s produce hard $S$ predictions via argmax of output logits and Stage 2 is identical across all conditions.

### 3.4.2 Partial Demographic Supervision

The original paper claims the method operates without demographic information, yet Stage 1 must predict an $S$ to produce a useful proxy. We investigate this tension directly by training each Stage 1 proxy with varying fractions of sensitive labels observed during training: $\rho \in \{0\%, 20\%, 50\%, 100\%\}$.

At ratio $\rho$, only a random subset of $\rho \cdot |V_{\text{train}}|$ training nodes have their sensitive label revealed to Stage 1. The subset is drawn independently per seed to ensure variance across seeds reflects label-subset uncertainty.

At $\rho = 0\%$, the `SClassifier` and `LatentClassifierVGAE` receive no $S$ supervision and save random-initialisation checkpoints essentially serving as zero-feature-information baselines. The VGAE at $\rho = 0\%$ trains without the $S$ reconstruction term.

### 3.5 Computational Requirements

All experiments were conducted on a single NVIDIA GeForce RTX 4090 GPU, completing a total of 840 runs across the entire configuration sweep in approximately 4 hours (see Appendix H for details).

## 4 Results

### 4.1 Reproduction

Table 4.1 compares our implementation against those reported by Wang et al. (2025b) for the three original datasets without demographic supervision ($\rho = 0\%$), which is the condition closest to the setup described in Wang et al. (2025b).

| Dataset | Metric | GCN | | | FairKD | | | fairGNN-WOD | | |
|---|---|---|---|---|---|---|---|---|---|---|
| | | Ours | Original Paper | $\Delta$ | Ours | Original Paper | $\Delta$ | Ours | Original Paper | $\Delta$ |
| **Credit** | Accuracy | 0.786 | 0.781 | $+0.005\uparrow$ | 0.785 | 0.711 | $+0.074\uparrow$ | **0.804** | 0.754 | $+0.050\uparrow$ |
| | F1 | 0.877 | 0.868 | $+0.009\uparrow$ | 0.877 | 0.796 | $+0.081\uparrow$ | **0.883** | 0.861 | $+0.022\uparrow$ |
| | SPD $\downarrow$ | 0.009 | 0.117 | $-0.108\uparrow$ | **0.008** | 0.094 | $-0.086\uparrow$ | 0.020 | 0.036 | $-0.016\uparrow$ |
| | EOD $\downarrow$ | 0.008 | 0.096 | $-0.088\uparrow$ | **0.007** | 0.075 | $-0.068\uparrow$ | 0.008 | 0.027 | $-0.019\uparrow$ |
| **Pokec-z** | AUC | **0.763** | 0.699 | $+0.064\uparrow$ | **0.763** | 0.673 | $+0.090\uparrow$ | 0.743 | 0.703 | $+0.040\uparrow$ |
| | F1 | **0.721** | 0.622 | $+0.099\uparrow$ | 0.718 | 0.592 | $+0.126\uparrow$ | 0.702 | 0.621 | $+0.081\uparrow$ |
| | SPD $\downarrow$ | 0.044 | 0.075 | $-0.031\uparrow$ | **0.032** | 0.045 | $-0.013\uparrow$ | 0.039 | 0.028 | $+0.011\downarrow$ |
| | EOD $\downarrow$ | 0.031 | 0.062 | $-0.031\uparrow$ | **0.021** | 0.048 | $-0.027\uparrow$ | 0.044 | 0.029 | $+0.015\downarrow$ |
| **Pokec-n** | AUC | 0.729 | 0.689 | $+0.040\uparrow$ | **0.736** | 0.663 | $+0.073\uparrow$ | 0.707 | 0.691 | $+0.016\uparrow$ |
| | F1 | 0.664 | 0.631 | $+0.033\uparrow$ | **0.671** | 0.603 | $+0.068\uparrow$ | 0.654 | 0.626 | $+0.028\uparrow$ |
| | SPD $\downarrow$ | **0.011** | 0.084 | $-0.073\uparrow$ | 0.013 | 0.067 | $-0.054\uparrow$ | 0.025 | 0.028 | $-0.003\uparrow$ |
| | EOD $\downarrow$ | **0.012** | 0.078 | $-0.066\uparrow$ | 0.014 | 0.064 | $-0.050\uparrow$ | 0.023 | 0.038 | $-0.015\uparrow$ |

Table 3: Experimental results comparison between our implementation without demographics ($\rho = 0.0$) and values of the original paper. Results are averaged across 5 random seeds; confidence intervals are narrow and omitted here for clarity (variance details can be found in Appendix B). $\Delta$ represents the performance difference (Ours $-$ Paper) — an upward arrow represents an improvement, a downward arrow a worse result. Best overall results per row are highlighted in bold.

**Utility results were partially reproduced.** GCN and FairKD achieve comparable or higher utility than is reported in the original paper across all datasets. Notably, in our experiments, FairKD matches the utility of GCN, whereas Wang et al. (2025b) report a slight decline in utility for the Credit dataset. fairGNN-WOD achieves the lowest AUC and F1 scores for the Pokec-z and Pokec-n datasets, which means we are unable to verify the claim that fairGNN-WOD maintains comparable utility to the GCN baseline. Interestingly, fairGNN-WOD does achieve the highest accuracy (0.804) and F1 score (0.883) for the Credit

dataset, although it is the least fair of all models (SPD = 0.020) for this dataset. This suggests that rather than suppressing demographic cues, fairGNN-WOD learns to exploit these as useful task-relevant signals.

**Baselines are more fair than originally reported.** While the original paper reports SPD values of 0.117, 0.075, and 0.084 for GCN for the Credit, Pokec-z and Pokec-n datasets, respectively, our GCN achieves substantially lower values of 0.009, 0.044 and 0.011. This is likely because of the low correlation of demographics with the prediction task labels in this dataset, as is shown in Table E. Ultimately, we are unable to reproduce the SPD score of 0.117 for the GCN baseline reported by Wang et al. (2025b) for the Credit dataset, even with alternative data splitting (see Appendix E.1).

**fairGNN-WOD does not consistently improve fairness over baselines.** For the Credit dataset, the SPD score for fairGNN-WOD (0.020) exceed both GCN (0.009) and FairKD (0.008), making baselines more fair in our experiments. For the Pokec-z dataset, the SPD score fairGNN-WOD (0.039) offers a marginal improvement over that of GCN (0.044) but is outperformed by FairKD (0.032). For SPD scores for the Pokec-n dataset, fairGNN-WOD (0.025) is outperformed by both GCN (0.011) and FairKD (0.013). The value of $\rho$ (0% vs 100%) makes a negligible difference for these outcomes, which is a finding we investigate further in Appendix F.

**Stage 1 produces accurate predictions for $S$ with minimal supervision.** As is shown in Appendix G, all Stage 1 proxies where $\rho \geq 20\%$ achieve comparable accuracy for predicting $S$. Moreover, the simple discriminative `SClassifier` matches or exceeds the VGAE at $\rho = 20\%$ on four of six datasets, casting doubt on the necessity of the stage-1 VGAE proposed by Wang et al. (2025b).

**Stage 1 proxy has no measurable impact.** Across all conditions, the three stage-1 proxy types and the Random Noise baseline produce fairness and utility that overlap within one standard deviation, on all six datasets and all four supervision levels (full results visualized in Figure F.2, Appendix F). Additionally, stage-1 performance has a negligible downstream effect on fairness (see Appendix F), meaning that stage-1 of the framework is fact entirely obsolete in our experiments.

### 4.2 Results on Additional Datasets

Table 4.2 reports results on the three additional datasets at $\rho = 0\%$ and full results across supervision ratios are in Appendix F. The pattern mirrors the reproduction findings: fairGNN-WOD improves fairness only when considering SPD on the NBA dataset, which is the only dataset with a meaningful $S$–$Y$ correlation (Table E), although variance is high. For the Bail dataset, fairGNN-WOD achieves the highest utility and lowest SPD, trading fairness for utility. For the DBLP dataset, all models perform comparably.

| Dataset | Metric | GCN | FairKD | fairGNN-WOD |
|---|---|---|---|---|
| Bail | Accuracy | 0.863±0.010 | 0.856±0.005 | **0.896**±0.026 |
| | F1 | 0.813±0.013 | 0.805±0.006 | **0.863**±0.033 |
| | SPD ↓ | **0.036**±0.006 | 0.037±0.004 | 0.061±0.008 |
| | EOD ↓ | **0.011**±0.010 | 0.016±0.005 | 0.016±0.009 |
| NBA | Accuracy | 0.681±0.024 | 0.694±0.017 | **0.727**±0.017 |
| | F1 | 0.720±0.018 | 0.735±0.012 | **0.755**±0.034 |
| | SPD ↓ | 0.142±0.031 | 0.168±0.053 | **0.094**±0.049 |
| | EOD ↓ | **0.147**±0.030 | 0.187±0.018 | 0.180±0.139 |
| DBLP | Accuracy | 0.944±0.005 | **0.947**±0.004 | 0.932±0.007 |
| | F1 | 0.917±0.008 | **0.921**±0.006 | 0.894±0.010 |
| | SPD ↓ | 0.050±0.012 | 0.070±0.017 | **0.042**±0.021 |
| | EOD ↓ | **0.018**±0.014 | 0.040±0.004 | 0.037±0.033 |

Table 4: Results on additional datasets at $\rho = 0\%$ (no demographic supervision for Stage 1). Mean ± std across 5 seeds. Best result per row are in **bold**.

## 5 Discussion

Our experiments yield four main findings, which we present below alongside their implications for the broader problem of reproducing complex fair-ML systems.

**Stage 1 has no measurable effect.**  The stage-1 architecture and the degree of demographic supervision, and consequently the performance of stage-1, has no measurable effect on downstream fairness or utility performance. All stage-1 variants, including the pure random-noise proxy, yield results that overlap within 1 standard deviation. This means that we are unable to verify the originally claimed contribution of the elaborate generative modeling of Stage 1, which is a central component of the fairGNN-WOD framework.

**Fairness improvement mainly depends on dataset $S$–$Y$ correlations.**  Across all six datasets, fairGNN-WOD shows a reduction in SPD only on NBA — the single dataset with a substantial $S$–$Y$ correlation (Table E) — where SPD drops from 0.142 (GCN) to 0.094 at $\rho = 0\%$. However, where the standard deviation on NBA is large, the improvement is not consistent across supervision levels, and disappears entirely for the EOD score, which makes this evidence unconvincing. In settings where baselines are already quite fair, fairGNN-WOD achieves comparable fairness performance or worse performance, trading in fairness for utility, as is the case for the Credit and Bail datasets. In general, a method evaluated only on datasets where $S$–$Y$ correlation is high may appear effective, but the same method on a near-independent dataset has nothing to correct. This makes the choice of dataset and split a first-order determinant of the reported result, and one that the original paper does not document.

**On reproducibility of the baseline disparity.**  We were unable to reproduce the reported GCN baseline SPD (e.g. 0.117 on Credit) under any standard or deliberately adversarial split with a large $Y$–$S$ correlation (Appendix E.1). On the Credit dataset as publicly available, $S$ and $Y$ are too weakly correlated for a simple GCN to exhibit the reported disparity while matching the original paper's utility metric. While this does not necessarily point to an error in the original work, it does mean the reported results cannot be reproduced from the dataset and information provided, and that the baseline against which the method's improvement is measured is itself not reconstructible.

**Ambiguous original description makes reproduction difficult.**  The fairGNN-WOD pipeline couples six interacting loss terms ($\mathcal{L}_P$, $\mathcal{L}_I$, $\mathcal{L}_D$, $\mathcal{L}_F$, the ELBO, and the HGR penalty) across two adversarially trained stages. A few unspecified details may be manageable in isolation, but in an adversarial multi-loss setting small differences in initialization, loss weighting, or architecture can significantly change behavior. We summarize the most significant ambiguities in the original paper in Appendix A, but many smaller ambiguities left out of the summary could just as well result in significant discrepancies between the original results and ours, since adversarial objectives are sensitive to the initial setup. For fairGNN-WOD, the original descriptions by Wang et al. (2025b) proved too ambiguous, resulting in failure to reproduce most major claims made in the original paper. Additionally, given the complexity of the proposed framework, the availability of the original code base is likely a necessary condition for complete reproduction of the original result.

**Communication with the original authors.**  To ensure an exact replication of the original setup, we contacted the authors on January 6, 2026, to request access to the source code. The authors declined to share the codebase but offered to answer specific clarification questions. Consequently, we submitted a list of inquiries on January 12, 2026, regarding conceptual ambiguities and hyperparameter specifications. As of this writing, no response to these questions has been received.

## 6   Conclusion

We reproduced fairGNN-WOD from scratch under substantial specification ambiguity. We matched the reported utility but could not reproduce the reported fairness improvements, tracing this to baselines that are already near-fair under a principled split and to a reported baseline disparity we could not reconstruct under any tested configuration. Our extensions show that the generative Stage-1 VGAE is functionally interchangeable with a simple classifier or even random noise. We conclude that the reported results are not reproducible from the descriptions provided. More broadly, our study illustrates how reproducibility of complex adversarial systems depends critically on detailed descriptions of the experimental setup and, ideally, on the availability of code, and emphasizes the importance of clear documentation.

## Acknowledgements

Acknowledgements will be added in the final version.

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

# A   Ambiguities in original setup

The algorithmic descriptions in Wang et al. (2025b) contain critical ambiguities. As a result, replicating their exact experimental setup is inherently challenging, and our implementations likely deviate from the original unreleased source code. Below, we analyse the most significant ambiguities in the original paper and justify our resolutions.

## A.1   Stage 1

A key feature of fairGNN-WOD (Wang et al., 2025b) is the inference of sensitive demographics $S$ by Stage 1 of the framework, which is achieved by modelling $S$ as a latent space of a VGAE. However, the mathematical formulation of this latent distribution is completely omitted. Given the datasets used in the original work contain discrete demographics, a relaxed categorical distribution seems most likely. However, a binary $S$ introduces a rather tight information bottleneck. Additionally, a related work tackling a similar problem with the same first author (Wang et al., 2025a) uses a Gaussian latent space.

The dimensionality of $S$ is also unspecified. A dimensionality of one is most consistent with the descriptions by Wang et al. (2025b), but since an entire graph is reconstructed from latent space $S$, this seems like a severe information bottleneck. Because of this we chose $C = 2$ dimensions — one per demographic group — and parameterise $q_\psi(S|Z)$ as a relaxed categorical distribution using the Gumbel-softmax estimator (Jang et al., 2016), which produces a probability vector $\mathbf{s} \in \Delta^{C-1}$ over the $C$ groups via:

$$\mathbf{s} = \mathrm{softmax}\left(\frac{\mathbf{z}_S + \mathbf{g}}{\tau}\right), \quad \mathbf{g} \sim \mathrm{Gumbel}(0,1)^C \tag{A.1}$$

where $\tau$ is a temperature parameter annealed towards zero during training. The hard demographic prediction is recovered as $\hat{s} = \arg\max_c s_c$. Note that for binary demographics ($C = 2$), this is equivalent to a scalar Bernoulli probability since $s_0 = 1 - s_1$, leaving a single free parameter despite the two-dimensional parameterisation. Additionally, modelling $S$ as a $C$-simplex rather than a scalar makes the reconstruction task more expressive while remaining interpretable as a demographic prediction.

The ELBO in Equation 2 contains a prior term $P(S|Z)$ that is never described in the original paper. One interpretation is that Stage 1 does not use a prior for $S$, and the term is an oversight or an abstract expression of $S$ as following from $Z$. If $q_\psi(S|Z)$ is parameterised as a categorical distribution, $P(S|Z)$ could be a uniform (or learned) or a unit prior. We interpret this as an unconditional prior over $S$ (the conditioning on $Z$ likely being a notational inconsistency) and implement it as a categorical distribution: uniform at $\rho = 0\%$, and set to the empirical $S$ distribution observed in the labelled training subset at $\rho > 0\%$.

The paper also overloads notation for input and reconstructed graphs $(A, X)$ and $(A', X')$, leaving the decoder ordering ambiguous. We implement the two decoders in parallel, conditioning both on the original $A$ rather than the reconstructed $A'$, since using a noisier reconstruction as input to the feature decoder would only degrade performance. Similarly, they do not differentiate between $S$ as sensitive attribute and $S$ as a latent space. Because of this we interpreted $S$ to serve as a latent space directly.

Finally, the two interconnected networks estimating $p_Z$ and $p_Y$ for the HGR term are mentioned but not described. We implement these as a single shared two-layer MLP with a split output head, which is the simplest interpretation consistent with the description.

## A.2   Stage 2

Stage 2 of the fairGNN-WOD framework (Wang et al., 2025b) consists of multiple components. However, two key components are described at a high level without concrete specification. The paper does not specify how per-channel discriminator predictions are aggregated into a channel sensitivity score, nor in what sense the masking algorithm is learnable. Because it appears to be the most straightforward way of extracting demographic data leak, we tie channel sensitivity inversely to the discriminator's confusion on that channel. We then mask proportionally to a channel's sensitivity score, as only masking sensitive channels based on a

threshold would not be fully learnable. The details of our implementations of both modules are described in Section 3.1.2.

## A.3 Parameters

Wang et al. (2025b) provide no explicit parameter values anywhere in the paper, appendices or a publicly available codebase. The sensitivity analysis in Figure 4 sweeps $\alpha$ and $\beta$ over a range but does not indicate which values were used to produce the results in Table 1. The number of channels $N_c$, the HGR weight $\lambda$, all layer sizes, encoder and decoder depths, latent dimensionalities, activation functions, learning rate, weight decay, and training schedule are entirely unspecified. Our choices, which simply follow reasonable standards of the field, are detailed in Appendix C.

# B Full Experimental Results

## B.1 Original Paper Results

Table B.1 lists Table 1 from Wang et al. (2025b) verbatim. All numbers in the reproduction and discussion sections are sourced from this table.

| | | GCN | GIN | FairKD | KSMOTE | FairRF | Reckoner | fairGNN-WOD |
|---|---|---|---|---|---|---|---|---|
| **Credit** | Acc ↑ | $0.781 \pm 0.016$ | $\mathbf{0.787 \pm 0.018}$ | $0.711 \pm 0.012$ | $0.736 \pm 0.009$ | $0.735 \pm 0.007$ | $0.736 \pm 0.021$ | $\underline{0.754 \pm 0.052}$ |
| | F1 ↑ | $0.868 \pm 0.023$ | $\mathbf{0.877 \pm 0.018}$ | $0.796 \pm 0.023$ | $0.817 \pm 0.012$ | $0.809 \pm 0.022$ | $0.817 \pm 0.015$ | $\underline{0.861 \pm 0.018}$ |
| | SPD ↓ | $0.117 \pm 0.013$ | $0.106 \pm 0.011$ | $0.094 \pm 0.036$ | $0.071 \pm 0.003$ | $0.067 \pm 0.017$ | $\underline{0.068 \pm 0.017}$ | $\mathbf{0.036 \pm 0.015}$ |
| | EOD ↓ | $0.096 \pm 0.017$ | $0.088 \pm 0.013$ | $0.075 \pm 0.042$ | $0.055 \pm 0.013$ | $0.057 \pm 0.018$ | $\underline{0.055 \pm 0.014}$ | $\mathbf{0.027 \pm 0.013}$ |
| **Pokec-z** | AUC ↑ | $0.699 \pm 0.024$ | $0.691 \pm 0.015$ | $0.673 \pm 0.021$ | $\underline{0.697 \pm 0.024}$ | $0.690 \pm 0.014$ | $0.692 \pm 0.020$ | $\mathbf{0.703 \pm 0.041}$ |
| | F1 ↑ | $\mathbf{0.622 \pm 0.012}$ | $0.613 \pm 0.007$ | $0.592 \pm 0.013$ | $0.611 \pm 0.018$ | $\underline{0.617 \pm 0.019}$ | $0.603 \pm 0.021$ | $0.621 \pm 0.032$ |
| | SPD ↓ | $0.075 \pm 0.025$ | $0.061 \pm 0.014$ | $0.045 \pm 0.014$ | $0.037 \pm 0.017$ | $\underline{0.032 \pm 0.012}$ | $0.036 \pm 0.018$ | $\mathbf{0.028 \pm 0.013}$ |
| | EOD ↓ | $0.062 \pm 0.013$ | $0.057 \pm 0.007$ | $0.048 \pm 0.009$ | $0.039 \pm 0.010$ | $\underline{0.034 \pm 0.012}$ | $0.033 \pm 0.010$ | $\mathbf{0.029 \pm 0.015}$ |
| **Pokec-n** | AUC ↑ | $0.689 \pm 0.015$ | $0.685 \pm 0.018$ | $0.663 \pm 0.016$ | $0.669 \pm 0.013$ | $0.673 \pm 0.013$ | $\underline{0.675 \pm 0.028}$ | $\mathbf{0.691 \pm 0.024}$ |
| | F1 ↑ | $\mathbf{0.631 \pm 0.022}$ | $0.629 \pm 0.008$ | $0.603 \pm 0.023$ | $0.611 \pm 0.018$ | $0.616 \pm 0.032$ | $\underline{0.619 \pm 0.032}$ | $0.626 \pm 0.029$ |
| | SPD ↓ | $0.084 \pm 0.013$ | $0.078 \pm 0.017$ | $0.067 \pm 0.015$ | $0.061 \pm 0.005$ | $0.056 \pm 0.027$ | $\underline{0.042 \pm 0.008}$ | $\mathbf{0.028 \pm 0.013}$ |
| | EOD ↓ | $0.078 \pm 0.019$ | $0.071 \pm 0.027$ | $0.064 \pm 0.013$ | $0.066 \pm 0.013$ | $0.061 \pm 0.016$ | $\underline{0.052 \pm 0.011}$ | $\mathbf{0.038 \pm 0.014}$ |

Table B.1: Listed from Wang et al. (2025b), Table 1. Comparison of fairGNN-WOD with baseline methods. Best result per row in **bold**, runner-up underlined.

## B.2 Our Full Results with No Demographic Supervision.

Table B.2 reports complete results for all six datasets at $\rho = 0\%$ (no demographic supervision to Stage 1), averaged over five seeds. FairGNN-WOD uses the VGAE proxy. Utility is Accuracy for Credit, Bail, NBA, DBLP and AUC for Pokec-z/n. For results of fairGNN-WOD at other demographic supervision rates see Appendix F.

# C Hyperparameter Details

Table C details the hyperparameter settings used. The weight for the fairness loss $\gamma$ was chosen to be 5.0 to match the magnitude of this loss with the other loss components of fairGNN-WOD.

| Dataset | Metric | GCN | FairKD | fairGNN-WOD |
|---|---|---|---|---|
| Credit | Accuracy | $0.786 \pm 0.001$ | $0.785 \pm 0.001$ | $\mathbf{0.804 \pm 0.002}$ |
| | F1 | $0.877 \pm 0.000$ | $0.877 \pm 0.000$ | $\mathbf{0.883 \pm 0.001}$ |
| | SPD ↓ | $\mathbf{0.009 \pm 0.003}$ | $\mathbf{0.008 \pm 0.002}$ | $0.020 \pm 0.004$ |
| | EOD ↓ | $0.008 \pm 0.003$ | $\mathbf{0.007 \pm 0.002}$ | $0.008 \pm 0.005$ |
| Pokec-z | AUC | $\mathbf{0.763 \pm 0.004}$ | $\mathbf{0.763 \pm 0.002}$ | $0.743 \pm 0.009$ |
| | F1 | $\mathbf{0.721 \pm 0.004}$ | $0.718 \pm 0.005$ | $0.702 \pm 0.010$ |
| | SPD ↓ | $0.044 \pm 0.005$ | $\mathbf{0.032 \pm 0.007}$ | $0.039 \pm 0.026$ |
| | EOD ↓ | $0.031 \pm 0.007$ | $\mathbf{0.021 \pm 0.009}$ | $0.044 \pm 0.010$ |
| Pokec-n | AUC | $0.729 \pm 0.013$ | $\mathbf{0.736 \pm 0.005}$ | $0.707 \pm 0.011$ |
| | F1 | $0.664 \pm 0.010$ | $\mathbf{0.671 \pm 0.005}$ | $0.654 \pm 0.007$ |
| | SPD ↓ | $\mathbf{0.011 \pm 0.008}$ | $0.013 \pm 0.008$ | $0.025 \pm 0.018$ |
| | EOD ↓ | $\mathbf{0.012 \pm 0.006}$ | $0.014 \pm 0.006$ | $0.023 \pm 0.019$ |
| Bail | Accuracy | $0.863 \pm 0.010$ | $0.856 \pm 0.005$ | $\mathbf{0.896 \pm 0.026}$ |
| | F1 | $0.813 \pm 0.013$ | $0.805 \pm 0.006$ | $\mathbf{0.863 \pm 0.033}$ |
| | SPD ↓ | $\mathbf{0.036 \pm 0.006}$ | $0.037 \pm 0.004$ | $0.061 \pm 0.008$ |
| | EOD ↓ | $\mathbf{0.011 \pm 0.010}$ | $0.016 \pm 0.005$ | $0.016 \pm 0.009$ |
| NBA | Accuracy | $0.681 \pm 0.024$ | $0.694 \pm 0.017$ | $\mathbf{0.727 \pm 0.017}$ |
| | F1 | $0.720 \pm 0.018$ | $0.735 \pm 0.012$ | $\mathbf{0.755 \pm 0.034}$ |
| | SPD ↓ | $0.142 \pm 0.031$ | $0.168 \pm 0.053$ | $\mathbf{0.094 \pm 0.049}$ |
| | EOD ↓ | $\mathbf{0.147 \pm 0.030}$ | $0.187 \pm 0.018$ | $0.180 \pm 0.139$ |
| DBLP | Accuracy | $0.944 \pm 0.005$ | $\mathbf{0.947 \pm 0.004}$ | $0.932 \pm 0.007$ |
| | F1 | $0.917 \pm 0.008$ | $\mathbf{0.921 \pm 0.006}$ | $0.894 \pm 0.010$ |
| | SPD ↓ | $0.050 \pm 0.012$ | $0.070 \pm 0.017$ | $\mathbf{0.042 \pm 0.021}$ |
| | EOD ↓ | $\mathbf{0.018 \pm 0.014}$ | $0.040 \pm 0.004$ | $0.037 \pm 0.033$ |

Table B.2: Full results at $\rho = 0\%$. Mean $\pm$ std across 5 seeds. Best result per row in **bold**.

| Model | Bail | Credit | DBLP | NBA | Pokec-n | Pokec-z |
|---|---|---|---|---|---|---|
| GCN | 5K | 5K | 166K | 10K | 21K | 22K |
| FairKD | 11K | 10K | 333K | 21K | 43K | 44K |
| SClassifier (Stage 1) | 5K | 5K | 166K | 10K | 21K | 22K |
| VGAE (Stage 1) | 93K | 92K | 1,383K | 134K | 221K | 227K |
| LatentClassifierVGAE (Stage 1) | 101K | 100K | 1,391K | 142K | 229K | 234K |
| HGR Estimator (Stage 1 aux) | 282K | 282K | 282K | 282K | 282K | 282K |
| fairGNN-WOD (Stage 2) | 145K | 143K | 1,754K | 195K | 304K | 311K |

Table C.5: Trainable parameter counts per dataset. The HGR estimator operates on the fixed latent space $Z \in \mathbb{R}^{32}$ and is input-size independent.

| Group | Parameter | Value | Note |
|---|---|---|---|
| | Learning rate | 0.01 | Adam optimiser |
| | Weight decay | 5e-4 | L2 regularisation |
| Shared | Max epochs | 200 | |
| | Early stopping patience | 20 | |
| | Warmup epochs | 50 | Early stopping disabled |
| GCN | Hidden size | 64 | |
| FairKD | KD temperature | 4 | |
| | KD loss weight | 0.5 | |
| | Encoder hidden size | 256 | |
| | Latent size $|Z|$ | 32 | |
| VGAE / | HGR hidden size | 512 | |
| LatentVGAE (Stage 1) | HGR steps per epoch | 10 | Inner maximisation steps |
| | $\lambda_{\text{HGR}}$ (max weight) | 0.5 | Schedule following Table C |
| | Hidden size | 64 | |
| | Channels $N_c$ | 8 | Disentangled latent factors |
| | $\alpha$ ($\mathcal{L}_I$ weight) | 1.0 | |
| fairGNN-WOD (Stage 2) | $\beta$ ($\mathcal{L}_D$ weight) | 1.0 | |
| | $\gamma$ ($\mathcal{L}_F$ weight) | 5.0 | Scaled to $\mathcal{L}_P$ magnitude |
| | $\lambda_{\text{DD}}$ | 0.1 | $\mathcal{L}_D$ Propagation to encoder |
| | Adversary steps per epoch | 2 | Inner discriminator steps |

Table C.3: Hyperparameter configurations for all models. Parameters shared across all models are listed in the top block. Model-specific parameters follow in separate blocks.

| Model | Term | Schedule | Start epoch | End epoch | Value range |
|---|---|---|---|---|---|
| VGAE / | KL weight $\beta$ | Sigmoid | $0.1W$ | $W$ | $0 \rightarrow 0.1$ |
| LatentVGAE | $\lambda_{\text{HGR}}$ | Linear | $0.5W$ | $W$ | $0 \rightarrow 0.5$ |
| | Gumbel $\tau$ | Linear | $0$ | $T$ | $1.0 \rightarrow 0.1$ |
| | $\mathcal{L}_F$ weight $\gamma$ | Linear | $0.2W$ | $0.6W$ | $0 \rightarrow 5.0$ |
| | $\mathcal{L}_D$ weight $\beta$ | Linear | $0.2W$ | $W$ | $0 \rightarrow 1.0$ |
| fairGNN-WOD | $\mathcal{L}_I$ weight $\alpha$ | Linear | $0.4W$ | $W$ | $0 \rightarrow 1.0$ |
| | $\lambda_{\text{DD}}$ | Sigmoid | $0.6W$ | $W$ | $0 \rightarrow 0.1$ |
| | Masking | Step | $0.2T$ | $0.2T$ | off $\rightarrow$ on |

Table C.4: Dynamic loss weight schedules. $W$ denotes the number of warmup epochs, $T$ shows max epochs (see Table C).

# D    Model Framework

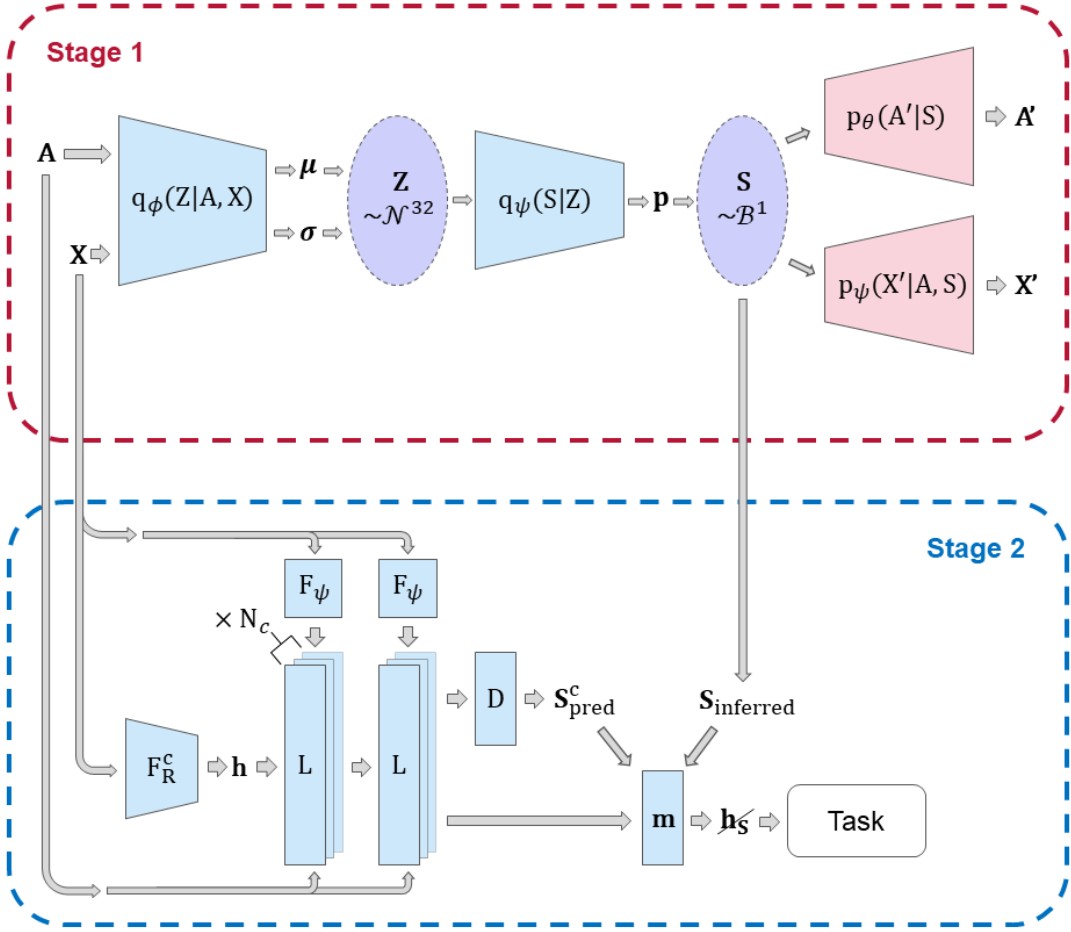

Figure D.1: Depiction of our implementation of the two-stage model framework proposed by Wang et al. (2025b). Components in red are only used during training, to compute a loss. Table D.6 provides a legend describing each component.

| Legend of Figure D.1 - Stage 1 (left) and Stage 2 (right) | | | |
|---|---|---|---|
| Name | Description | Name | Description |
| $q_\phi(Z\|A, X)$: | Encoder modeling $P(Z\|A, X)$ | $F_\psi$ | Adaptive assigner |
| $q_\psi(S\|Z)$: | Encoder modeling $P(S\|Z)$ | $F_c^R$ | Obtains initial hidden representations |
| $p_\theta(A'\|S)$: | Decoder modeling $P(A'\|S)$ | $L$ | Graph convolutional layer |
| $p_\eta(X'\|A, S)$: | Decoder modeling $P(X'\|A, S)$ | $m$ | Learnable masking algorithm |
| $A$: | Input adjacency matrix | $D$ | Discriminator |
| $X$: | Input node features | $h$ | Hidden node representation |
| $A'$: | Predicted adjacency matrix | $h_S$ | Final masked node representation |
| $X'$: | Predicted node features | $S_{pred}^c$ | Demographics prediction per channel |
| $\mu$: | Mean of $Z$ | $S_{inferred}$ | Inferred demographics label |
| $\sigma$: | Standard deviation of $Z$ | $N_c$ | Number of channels in GNN |
| $p$: | Probability of $S$ | | |

Table D.6: Legend of Figure D.1.

# E    Dataset Statistics and Split Analysis

Table E shows the correlations between sensitive attribute and classification label in all datasets.

| Dataset | $P(S{=}1 \mid Y{=}0)$ | $P(S{=}1 \mid Y{=}1)$ | Gap |
|---|---|---|---|
| Credit | 10.9% | 8.6% | 2.3% |
| Pokec-z | 32.2% | 36.4% | 4.2% |
| Pokec-n | 32.7% | 33.4% | 0.7% |
| Bail | 53.4% | 46.5% | 6.9% |
| NBA | 38.5% | 25.0% | 13.5% |
| DBLP | 17.0% | 18.8% | 1.8% |

Table E.7: Sensitive attribute correlation with target label in the test split. $P(S{=}1 \mid Y{=}k)$ is the fraction of nodes with $S{=}1$ within each target class. A small gap between $Y{=}0$ and $Y{=}1$ indicates low $S$–$Y$ correlation, which produces near-zero SPD for any classifier regardless of fairness constraints.

## E.1    Effect of Split Strategy on Baseline Fairness

The paper reports a GCN SPD of $0.117 \pm 0.013$ on Credit, while our implementation yields $0.009 \pm 0.003$ under a standard stratified split. We conducted two experiments to determine whether this discrepancy could be explained by split methodology.

**Non-stratified split.** Replacing our stratified 50/25/25 split with a random non-stratified split produced SPD $\approx 0.019$ — approximately double our baseline but an order of magnitude below the paper's value and notably still lower than fairGNN-WOD despite a more favorable split.

**Maximally correlated split.** We constructed a training set with an artificial $S$–$Y$ correlation of 72.3% by selecting 90% of $(S{=}1, Y{=}1)$ and $(S{=}0, Y{=}0)$ nodes and filling only the remaining 10% of the misaligned groups. Even under this extreme distortion, GCN achieved a mean SPD of only $0.04 + -0.009$ — still well below 0.117. Additionally, the accuracy collapsed to $\approx 0.22$ due to the heavy test subset skew.

We conclude that the reported baseline SPD of 0.117 cannot be reproduced through any standard or artificially skewed data split on the Credit dataset as distributed. Credit's graph structure does not transmit sufficient demographic signal for a GCN to achieve this level of disparity while maintaining normal predictive accuracy. The paper's split strategy remains unspecified and unidentifiable from the information provided.

# F    fairGNN-WOD Sensitivity to Demographic Supervision

Table F reports fairGNN-WOD performance using the VGAE proxy across all supervision ratios $\rho$. Results are averaged over five seeds; standard deviations are shown in parentheses.

Figure F.2 visualises the same comparison across all six datasets and all three proxy types. The overwhelming pattern is flatness: across 17 of 18 dataset–metric combinations, the lines for VGAE, LatentVGAE, and SClassifier are statistically indistinguishable from each other and from the random noise baseline regardless of $\rho$. This confirms that neither the choice of Stage 1 architecture nor the amount of demographic supervision has any meaningful effect on downstream fairness or utility.

# G    Stage 1 Sensitive Attribute Prediction Accuracy

Table G reports the validation accuracy of each Stage 1 proxy in predicting the sensitive attribute $S$. At $\rho = 0\%$, VGAE and LatentVGAE receive no $S$ supervision and collapse to near-chance accuracy on most datasets. At $\rho = 20\%$, all models recover meaningful predictive accuracy within 5–15 points of their full-supervision performance, with further gains from additional supervision being marginal. Bail is a consistent

| Dataset | Metric | $\rho = 0\%$ | $\rho = 20\%$ | $\rho = 50\%$ | $\rho = 100\%$ |
|---------|--------|--------------|---------------|---------------|----------------|
| Credit | Utility | $0.804 \pm 0.002$ | $0.801 \pm 0.005$ | $0.803 \pm 0.003$ | $0.801 \pm 0.007$ |
|  | SPD | $0.020 \pm 0.004$ | $0.017 \pm 0.010$ | $0.021 \pm 0.003$ | $0.016 \pm 0.008$ |
|  | EOD | $0.008 \pm 0.005$ | $0.007 \pm 0.006$ | $0.010 \pm 0.004$ | $0.007 \pm 0.003$ |
| Pokec-z | Utility | $0.743 \pm 0.009$ | $0.748 \pm 0.012$ | $0.745 \pm 0.006$ | $0.752 \pm 0.020$ |
|  | SPD | $0.039 \pm 0.026$ | $0.032 \pm 0.011$ | $0.040 \pm 0.023$ | $0.036 \pm 0.027$ |
|  | EOD | $0.044 \pm 0.010$ | $0.027 \pm 0.010$ | $0.026 \pm 0.019$ | $0.027 \pm 0.026$ |
| Pokec-n | Utility | $0.707 \pm 0.011$ | $0.712 \pm 0.009$ | $0.708 \pm 0.016$ | $0.709 \pm 0.009$ |
|  | SPD | $0.025 \pm 0.018$ | $0.016 \pm 0.009$ | $0.016 \pm 0.012$ | $0.020 \pm 0.011$ |
|  | EOD | $0.023 \pm 0.019$ | $0.027 \pm 0.018$ | $0.018 \pm 0.010$ | $0.021 \pm 0.015$ |
| Bail | Utility | $0.896 \pm 0.026$ | $0.896 \pm 0.015$ | $0.897 \pm 0.012$ | $0.898 \pm 0.020$ |
|  | SPD | $0.061 \pm 0.008$ | $0.063 \pm 0.004$ | $0.064 \pm 0.004$ | $0.060 \pm 0.013$ |
|  | EOD | $0.016 \pm 0.009$ | $0.017 \pm 0.007$ | $0.016 \pm 0.009$ | $0.012 \pm 0.013$ |
| NBA | Utility | $0.727 \pm 0.017$ | $0.722 \pm 0.020$ | $0.762 \pm 0.033$ | $0.724 \pm 0.021$ |
|  | SPD | $0.094 \pm 0.049$ | $0.040 \pm 0.021$ | $0.081 \pm 0.094$ | $0.078 \pm 0.051$ |
|  | EOD | $0.180 \pm 0.139$ | $0.093 \pm 0.015$ | $0.153 \pm 0.126$ | $0.113 \pm 0.122$ |
| DBLP | Utility | $0.932 \pm 0.007$ | $0.933 \pm 0.009$ | $0.934 \pm 0.020$ | $0.933 \pm 0.021$ |
|  | SPD | $0.042 \pm 0.021$ | $0.063 \pm 0.007$ | $0.067 \pm 0.031$ | $0.066 \pm 0.039$ |
|  | EOD | $0.037 \pm 0.033$ | $0.015 \pm 0.014$ | $0.030 \pm 0.021$ | $0.040 \pm 0.024$ |

Table F.8: fairGNN-WOD (VGAE proxy) across demographic supervision ratios $\rho$. Utility is Accuracy for Credit, Bail, NBA, DBLP and AUC for Pokec-z/n.

exception: even at $\rho = 100\%$, no proxy exceeds 58% accuracy, indicating that race is not reliably recoverable from the graph structure and features of this dataset. The anomalously high VGAE accuracy on DBLP at $\rho = 0\%$ (0.854) suggests that gender leaks through the bag-of-words node features without requiring explicit supervision.

| Model | Dataset | $\rho = 0\%$ | $\rho = 20\%$ | $\rho = 50\%$ | $\rho = 100\%$ |
|-------|---------|--------------|---------------|---------------|----------------|
| VGAE | Bail | 0.498 | 0.498 | 0.499 | 0.501 |
|  | Credit | 0.582 | 0.775 | 0.836 | 0.895 |
|  | DBLP | 0.854 | 0.853 | 0.856 | 0.876 |
|  | NBA | 0.536 | 0.782 | 0.782 | 0.782 |
|  | Pokec-n | 0.663 | 0.849 | 0.814 | 0.820 |
|  | Pokec-z | 0.473 | 0.802 | 0.819 | 0.755 |
| LatentVGAE | Bail | 0.503 | 0.501 | 0.504 | 0.511 |
|  | Credit | 0.587 | 0.839 | 0.869 | 0.850 |
|  | DBLP | 0.715 | 0.817 | 0.868 | 0.862 |
|  | NBA | 0.744 | 0.754 | 0.810 | 0.787 |
|  | Pokec-n | 0.402 | 0.717 | 0.851 | 0.855 |
|  | Pokec-z | 0.528 | 0.720 | 0.733 | 0.733 |
| SClassifier | Bail | — | 0.549 | 0.567 | 0.577 |
|  | Credit | — | 0.778 | 0.724 | 0.713 |
|  | DBLP | — | 0.759 | 0.797 | 0.839 |
|  | NBA | — | 0.813 | 0.805 | 0.844 |
|  | Pokec-n | — | 0.854 | 0.876 | 0.896 |
|  | Pokec-z | — | 0.869 | 0.880 | 0.899 |

Table G.9: Stage 1 validation accuracy on $S$ prediction across supervision ratios. SClassifier has no $\rho = 0\%$ entry as it saves a random-initialisation checkpoint at zero supervision. Values are medians across seeds.

# H    Computational Requirements and Training Sweep

All experiments were executed on a single workstation equipped with 16 vCPUs (AMD Ryzen 9 7950X), 61 GB of system RAM, and an NVIDIA GeForce RTX 4090 (24 GB) GPU.

As detailed in Table H, we conducted a comprehensive configuration sweep across datasets, seeds, architectures, and visibility ratios. Baseline models (`GCN`, `FairKD`) and Stage 1 `SClassifier` require $\sim$ 1–5 seconds per run. Stage 1 `VGAE` training takes $\sim$10 seconds on large datasets (*Pokec_N*) and under 5 seconds on smaller ones (*NBA*, *DBLP*). Stage 2 training ranges from 3–50 seconds heavily dependent on the dataset.

| Dimension | Experimental Choices |
|---|---|
| Datasets | 6 distinct domains (Credit, NBA, DBLP, Pokec-z/n, Bail) |
| Random Seeds | 5 independent initialisations |
| Stage 1 Architectures | 4 variations (VGAE, `SClassifier`, `LatentClassifierVGAE`, `RandomNoise`) |
| Evaluated Models | 3 frameworks (GCN, FairKD, fairGNN-WOD) |
| Sensitive Visibility Ratios | 4 visible sensitive ratios (0%, 20%, 50%, 100%) |

Table H.10: Experimental Search Grid Dimensions

Accounting for baselines, the fact that each fairGNN-WOD evaluation requires a preceding Stage 1 pre-training execution and that fairGNN-WOD + `RandomNoise` only runs for one $\rho = 0.0$, the matrix yields exactly 840 total runs:

$$\text{Total Runs} = \underbrace{(6 \times 5 \times 2 \times 1 \times 1)}_{\text{Baselines (GCN, FairKD)}} + \underbrace{(6 \times 5 \times 3 \times 2 \times 4)}_{\text{fairGNN-WOD + Stage 1s}} + \underbrace{(6 \times 5 \times 1 \times 2 \times 1)}_{\text{fairGNN-WOD + Random S}} = 840 \qquad \text{(H.2)}$$

The entire evaluation was completed in approximately 4 hours of continuous computation (including I/O), averaging $\sim$ 25 seconds per run.

# I    Removed Experimental Runs

During evaluation, certain hyperparameter configurations can cause the model to suffer from total collapse (i.e., the model predicts a single majority class for all instances). When this occurs, fairness metrics like Statistical Parity Difference (SPD) and Equal Opportunity Difference (EOD) artificially drop to exactly 0.0, creating a false impression of perfect fairness.

To ensure honest reporting and prevent these unstable artifacts from skewing the final analysis, runs exhibiting a statistical parity difference below a threshold of $\epsilon = 10^{-4}$ were flagged as collapsed and pruned from the main results.

Out of 450 total experimental runs, 4 runs (0.89%) were pruned due to model collapse.

**Summary of Pruned Experimental Runs**

Table I details the specific configurations where the model collapsed (SPD $\approx$ 0 and EOD $\approx$ 0):

| Seed | Model | Dataset | Stage 1 | $\rho$ | SPD | EOD |
|---|---|---|---|---|---|---|
| 1 | fairGNN-WOD | nba | SClassifier | 0.2 | 0.0 | 0.0 |
| 789 | fairGNN-WOD | pokec_n | SClassifier | 0.2 | 0.0 | 0.0 |
| 789 | fairGNN-WOD | pokec_n | SClassifier | 0.5 | 0.0 | 0.0 |
| 789 | fairGNN-WOD | pokec_n | SClassifier | 1.0 | 0.0 | 0.0 |

Table I.11: Pruned experimental runs exhibiting model collapse.

As shown in Table I, the collapse behavior was highly isolated, appearing exclusively in fairGNN-WOD when utilizing the `SClassifier` stage, with a particular vulnerability on the `pokec_n` dataset's split on seed = 789 with at least some demographics information visible.

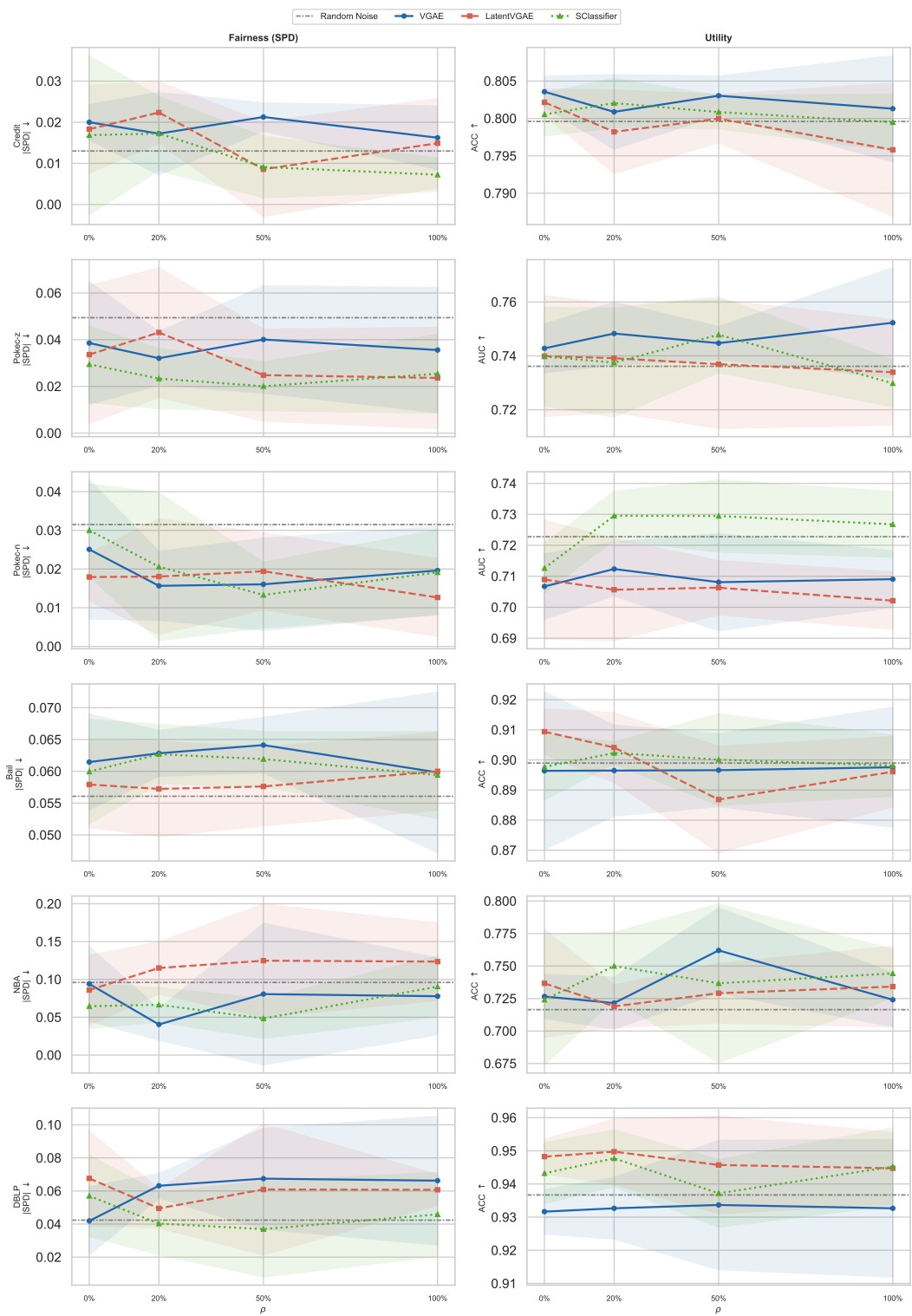

Figure F.2: FairGNN-WOD fairness (SPD) and utility across Stage 1 proxy types and demographic supervision ratios $\rho \in \{0\%, 20\%, 50\%, 100\%\}$, for all six datasets. Shaded bands show $\pm 1$ standard deviation across five seeds. The dashed grey line marks the random noise baseline. EOD follows the same pattern as SPD across all conditions and is omitted for brevity; full results including EOD are reported in Table F.

