# OpenReview forum: "Revisiting fairGNN-WOD: A Reproduction and Analysis of Fair Graph Learning Without Demographics"
_TMLR — Under review for TMLR_

### Review · Reviewer_KP8Y · 2026-06-22

**Summary Of Contributions:**

This paper reproduces and analyzes fairGNN-WOD, a two-stage method for fair graph learning without demographic information. Because the original code is unavailable and several implementation details are underspecified, the authors implement the method from scratch. They find that utility results are broadly reproducible, but the reported fairness gains are not: in their experiments, GCN and FairKD baselines are already much fairer than in the original report. They further test alternative Stage-1 proxies, different sensitive-label visibility ratios, and three additional datasets, concluding that the Stage-1 VGAE has little measurable downstream effect.

# Strengths

1. The topic is relevant: the paper probes a strong and practically important claim, namely fairness without demographic information.

2. The reproduction difficulties are documented concretely, especially in Appendix A, where the authors identify ambiguities in the original model, losses, masking, latent sensitive representation, and hyperparameters.

3. The extensions are useful. Testing simpler Stage-1 proxies, random noise, partial demographic supervision, and additional datasets helps assess whether the original method's mechanism is actually doing what it claims.

# Weaknesses

1. The main negative conclusions are too strong. The paper often moves from "our plausible implementation does not show an effect" to "Stage 1 is obsolete" or "random noise is interchangeable." Given the substantial implementation ambiguity, these claims should be qualified or supported with broader implementation and hyperparameter sensitivity.

2. The statistical evidence is limited. Most results use five seeds and mean +/- standard deviation, and several conclusions rely on visual overlap in Figure F.2. Claims such as "no measurable impact" would be stronger with paired tests, confidence intervals, effect sizes, or equivalence-style analysis.

3. Hyperparameter sensitivity is insufficient. fairGNN-WOD has many interacting loss weights, adversarial components, masking choices, and schedules, but the paper mainly reports one reasonable configuration. This makes it hard to know whether the negative result reflects the method or this implementation regime.

4. The explanation of baseline fairness is somewhat underdeveloped. Appendix E relates low SPD to weak S-Y correlation, but low S-Y correlation alone does not rule out S leakage through features or graph structure. The paper should add group-wise prediction rates and clearer diagnostics before making broad claims about why baselines are near-fair.

5. Some experimental reporting needs clarification. SPD/EOD definitions, thresholding, positive group/class conventions, and whether absolute values are used should be stated explicitly. Appendix I's removal of collapsed runs also needs clearer reporting, including results with and without pruning.

**Additional Comments:**

n/a

**Audience:**

Yes

**Audience Explanation:**

Although centered on one prior method, the paper addresses broader issues in fair graph learning: how to evaluate fairness without demographic supervision, how baseline disparity affects conclusions, and how underspecified adversarial systems hinder reproducibility. This is likely useful to a subset of TMLR readers.

**Broader Impact Concerns:**

The paper studies fairness with sensitive attributes such as race, inferred gender, nationality, and age. The authors should emphasize that these attributes are used for auditing and research evaluation, not deployment guidance. They should also avoid implying that fairness can be responsibly assessed without reliable demographic information; even if training does not use demographics, evaluation usually still requires careful group-level auditing.

**Claims And Evidence:**

Yes

**Claims Explanation:**

Partially. The paper supports the narrower claim that, under the authors' implementation and tested settings, fairGNN-WOD's reported fairness gains are not reproduced and Stage-1 proxy choice has little visible effect. It does not yet fully support stronger claims about the original method being ineffective or Stage 1 being generally unnecessary.

**Requested Changes:**

Critical to acceptance:

1. Qualify the strongest conclusions or add sensitivity analyses across plausible implementations and key hyperparameters.

2. Provide stronger statistical support for the main comparisons, especially the Stage-1 proxy ablations.

3. Clearly define SPD, EOD, thresholding, positive class/group conventions, and whether absolute values are used.

4. Report the impact of collapsed-run pruning and clarify the run counts in Appendices H and I.

5. Add diagnostics explaining why baselines are near-fair, beyond only S-Y correlation.

Would strengthen the paper:

1. Include more original baselines, or explicitly limit claims to GCN and FairKD.

2. Provide exact splits, seeds, and per-run metrics.

3. Add split-sensitivity analysis beyond Credit.

---

### Review · Reviewer_NKQU · 2026-06-24

**Summary Of Contributions:**

This paper revisits fairGNN-WOD, a recently proposed method for fair graph learning without demographic information. Since the original code is unavailable, the authors reimplement the method from scratch, evaluate it on the original datasets and three additional datasets, and report that they can reproduce utility but not the claimed fairness improvements. The paper further argues that the Stage-1 VGAE component has no measurable downstream contribution and that simpler proxies or even random noise lead to similar results.

**Additional Comments:**

NA

**Audience:**

No

**Audience Explanation:**

NA

**Claims And Evidence:**

No

**Claims Explanation:**

While the topic of reproducibility in fair graph learning is important, the current manuscript is not sufficiently mature for publication. The paper reads more like an incomplete reproduction report than a rigorous research contribution. The implementation contains many subjective design choices, the experimental protocol is not sufficiently justified, and the conclusions are often stronger than what the evidence supports.

1. The abstract is not written in a standard scholarly form. The abstract is overly direct and conclusion-driven, but does not clearly describe the problem, protocol, experimental setting, scope of reproduction, or main evidence in a balanced way. Statements such as “we fail to reproduce” and “Stage 1 has no measurable contribution” are strong claims, but the abstract does not sufficiently qualify them by emphasizing that the original code is unavailable and that the implementation is based on many assumptions. For a reproducibility paper, the abstract should be especially careful and neutral.

2. The central conclusion is not sufficiently supported because the implementation is highly uncertain. The authors repeatedly acknowledge that the original method description is ambiguous and that their implementation may deviate from the original method. However, the paper still draws strong conclusions such as Stage 1 being “obsolete” or having “no measurable impact.” This is logically problematic. If the implementation may differ substantially from the original method, then negative results may reflect implementation mismatch rather than failure of the original method.

3. The paper lacks a rigorous verification process for the reimplementation. Since the original code is unavailable, the burden is on the authors to demonstrate that their implementation is a faithful approximation. The manuscript lists many assumptions, but does not provide enough sanity checks, ablations, unit-level validation, or independent verification to show that each module behaves as intended. This is especially important for a complex two-stage adversarial framework with VGAE, HGR penalty, disentanglement, masking, MMD, and fairness losses.

4. Many key design choices appear arbitrary or insufficiently justified. The paper introduces specific architectural choices, hidden sizes, schedules, loss weights, proxy designs, Gumbel-softmax relaxation, sensitivity scoring, masking strategy, and training schedules. These choices may substantially affect fairness outcomes, yet the manuscript does not convincingly justify them or show that conclusions are robust to them. A reproduction study should separate unavoidable ambiguity from author-introduced design decisions.

5. The paper mainly relies on SPD and EOD, but does not provide a sufficiently deep fairness analysis across thresholds, calibration, subgroup performance, confidence intervals, statistical tests, or sensitivity to split construction. Since the key claim concerns fairness non-reproducibility, the evaluation should be much more rigorous. Reporting mean ± standard deviation over five seeds is not enough for such a strong conclusion.

6. The comparison with the original paper is potentially unfair. The manuscript compares its reproduced values against the original reported values, but the exact preprocessing, splits, model selection criteria, thresholding rules, and hyperparameters are unknown. Under such uncertainty, it is risky to claim that the original result is not reproducible. A more careful framing would be: “we could not reproduce the results under our best-effort implementation,” rather than implying that the original method itself is invalid.

**Requested Changes:**

See the comments above.

---

### Review · Reviewer_xMYS · 2026-07-13

**Summary Of Contributions:**

The authors reimplement fairGNN-WOD (Wang et al., IJCAI 2025) from scratch, since no code was released and the original authors declined to share it. They evaluate on the three original datasets plus three additional ones (Bail, NBA, DBLP), across five seeds and four demographic-supervision ratios ρ ∈ {0, 20, 50, 100}%, and release their code. Their baselines match or exceed the originally reported utility, but the reported fairness gains do not materialize: their GCN baseline is already near-fair (SPD 0.009 versus the reported 0.117 on Credit), and fairGNN-WOD does not consistently improve fairness over it. Substituting the Stage-1 VGAE with a simple classifier, a latent-classifier variant, or random noise leaves downstream fairness and utility indistinguishable within one standard deviation in 17 of 18 dataset–metric combinations. The authors attribute observed fairness effects to dataset-level S–Y correlation, and show through deliberately skewed splits that the original baseline disparity on Credit appears unreachable on the dataset as distributed.

**Audience:**

Yes

**Audience Explanation:**

I believe the work will ensure people don't pursue a research thread that seems promising form the original work, but primarily is not.

**Broader Impact Concerns:**

None. The datasets are standard public fairness benchmarks, and the work reduces the risk of the community overclaiming fairness without demographics.

**Claims And Evidence:**

Yes

**Claims Explanation:**

## Strengths

1. The reproduction is unusually transparent. Every ambiguity in the original description is catalogued with the chosen resolution and its rationale (Appendix A), code is released, and the communication attempts with the original authors are documented with dates.

2. The Stage-1 controls directly test the framework's causal claim rather than merely re-running it. The decisive control is that at ρ ≥ 20% every proxy predicts S accurately (75–90% validation accuracy, Table G.9) yet downstream fairness and utility are unchanged, so the null result cannot be explained by a failed proxy. A complementary observation strengthens the case: at ρ = 0% the VGAE's S accuracy collapses to near chance on five of six datasets, which independently undermines the original claim that the generative process recovers S without supervision.

3. The split analysis on Credit (Appendix E.1) converts "we could not reproduce a number" into "the number appears unreachable." Even a training set constructed with 72.3% S–Y correlation yields GCN SPD of only 0.04 (at collapsed accuracy), an order of magnitude below the reported 0.117. This is the paper's strongest evidence that the discrepancy is not an artifact of the reimplementation.

4. The S–Y correlation analysis (Table E.7) supplies a mechanism for the negative result: on datasets where S and Y are nearly independent, any classifier is near-fair and a fairness intervention has nothing to correct. This makes dataset and split choice a first-order determinant of reported fairness gains, a point of general value beyond this reproduction.

## Weaknesses

1. **A single untuned configuration limits the strength of the negative result.** The original paper specifies no hyperparameters, and the authors fix α = β = 1.0, γ = 5.0, λ_DD = 0.1 with hand-designed loss schedules and no search (Appendix C). Adversarial multi-loss systems are sensitive to these choices, so "we could not reproduce the fairness gains" may partly mean "we did not find the configuration in which they appear." The baseline-disparity finding is immune to this concern because it involves no fairGNN-WOD components, but the claim that Stage 1 has no measurable effect is not: under different Stage-2 loss weights, the pipeline might exploit the proxy.

2. **No statistical test supports the central no-effect claim.** Overlap within one standard deviation across five seeds is not a significance criterion, and a conclusion of "no measurable effect" calls for an equivalence-style analysis, or at minimum paired per-seed comparisons with confidence intervals, rather than visual overlap of ±1 SD bands (Figure F.2).

3. **The fairness-metric protocol is under-specified.** SPD and EOD require binarized predictions, but the paper does not state the threshold policy (fixed 0.5, validation-tuned, or otherwise), nor the binarization used on Pokec where utility is reported as AUC. Because thresholding can materially change SPD and EOD, the protocol must be stated and shown to be identical across methods for the comparisons to be auditable.

4. **Baseline coverage is narrower than the original's, and FairKD is never identified.** Only GCN and FairKD of the original's six baselines are rerun, and FairKD appears in every results table without a citation or a sentence of description, so the reader cannot even identify the method being compared.

5. **Acknowledged implementation deviations are not ablated.** MSE reconstruction for heterogeneous (largely binary or categorical) features, conditioning the feature decoder on A rather than A′, hard argmax S proxies rather than soft probabilities, and the particular HGR estimator are each reasonable, but none is varied, so their contribution to the discrepancy with the original results is unknown.

6. **Internal inconsistencies weaken the presentation.** The abstract states "we reproduce utility results" while Section 4.1 reports utility as only "partially reproduced" and declines to verify the comparable-utility claim on Pokec. Table cross-references do not match table numbers (e.g., "Table 4.1" for Table 3, "Table 3.2" for Table 1). Appendix H reports 840 total runs while Appendix I reports 450 without reconciling the two. Minor typos remain ("0.04 + −0.009" in E.1; "is fact entirely obsolete" in §4.1).

**Requested Changes:**

Critical to my recommendation:

1. Scope the headline claims to the reimplementation. Statements such as "we find no measurable contribution of stage 1 of the original framework" (abstract) should read as claims about the authors' implementation and configuration, since faithfulness to the unreleased original cannot be established.

2. Specify the thresholding and binarization protocol for SPD and EOD on every dataset, confirm it is identical across all methods and seeds, and surface it in the main text.

3. Support the no-effect claims with a statistical analysis appropriate to a null result (equivalence testing or paired per-seed comparisons with confidence intervals), in place of visual ±1 SD overlap.

4. Add a Stage-2 loss-weight sensitivity analysis (at minimum α, β, γ on one dataset where the original reports its largest gain), or explicitly bound the no-effect conclusion by the single configuration tested.

5. Cite and briefly describe FairKD, and reconcile the abstract's utility claim with Section 4.1.

Would strengthen the work:

6. Ablate hard versus soft S proxies in Stage 2, and the HGR term (removal, or targeting Ŝ–Y rather than Z–Y), to localize which components matter under the reimplementation.

7. Reconcile the 840 versus 450 run counts, repair table cross-references, and fix the typos noted above.

8. Situate the finding that well-tuned baselines close reported fairness gaps within the existing literature on baseline strength in fair ML, which would sharpen the discussion without new experiments.